# The Characteristics of Structural Properties and Diffusion Pathway of Alkali in Sodium Trisilicate: Nanoarchitectonics and Molecular Dynamic Simulation

**DOI:** 10.3390/ijms25115628

**Published:** 2024-05-22

**Authors:** Pham Huu Kien, Giap Thi Thuy Trang

**Affiliations:** Department of Physics, Thai Nguyen University of Education, No. 20 Luong Ngoc Quyen, Thai Nguyen 250000, Vietnam; kienph@tnue.edu.vn

**Keywords:** molecular dynamics simulation, nanoarchitectonics, Voronoi polyhedron, structure, diffusion pathway

## Abstract

Based on nanoarchitectonics and molecular dynamics simulations, we investigate the structural properties and diffusion pathway of Na atoms in sodium trisilicate over a wide temperature range. The structural and dynamics properties are analyzed through the radial distribution function (RDF), the Voronoi Si- and O-polyhedrons, the cluster function f_CL_(r), and the sets of fastest (SFA) and slowest atoms (SSA). The results indicate that Na atoms are not placed in Si-polyhedrons and bridging oxygen (BO) polyhedrons; instead, Na atoms are mainly placed in non-bridging oxygen (NBO) polyhedrons and free oxygen (FO) polyhedrons. Here BO, NBO, and FO represent O bonded with two, one, and no Si atoms, respectively. The simulation shows that O atoms in sodium trisilicate undergo numerous transformations: NBF0 ↔ NBF1, NBF1 ↔ NBF2, and BO0 ↔ BO1, where NBF is NBO or FO. The dynamics in sodium trisilicate are mainly distributed by the hopping and cooperative motion of Na atoms. We suppose that the diffusion pathway of Na atoms is realized via hopping Na atoms alone in BO-polyhedrons and the cooperative motion of a group of Na atoms in NBO- and FO-polyhedrons.

## 1. Introduction

It is well known that the structure of silica (SiO_2_) is an archetypal network-forming system containing SiO_4_ tetrahedra. The addition of doped Na ions generates non-bridging oxygen (NBO) [1,2]. Therefore, sodium trisilicate (Na_2_O∙3SiO_2_) has various anomalous properties which are essential for industrial applications, ceramics, metallurgy, and glass technologies, as well as for understanding the fundamentals of minerals [3,4,5]. Na_2_O∙3SiO_2_ has been extensively studied using experimental methods such as photoelectron spectroscopy [6], X-ray diffraction [4,7], in situ Raman spectroscopy, and elastic neutron scattering [8], along with various simulation techniques [9,10,11,12]. Nesbitt et al. have characterized two types of network oxygen in sodium silicate: three-fold-coordinated BO-Na and two-fold-coordinated Si-O-Si [6]. The doped Na ions function as a network modifier, causing significant alterations in the random network of corner-shared SiO_4_ tetrahedra with the formation of FO and NBO. Davidenko et al. reported that Na_2_O-SiO_2_ has a micro-heterogeneous structure which contains noncrystalline micro-groups such as silica, Na disilicate, and Na monosilicate [7]. Zhao et al. indicated that, when enough Na_2_O is added into the glass matrix to create two NBO atoms (Q2 species), the silica network loses its three-dimensional connectivity needed to sustain the local transformations between α-like and β-like rings, and glass softens upon heating due to the dominant anharmonic effect [8].

The nanoarchitectonics and molecular dynamics (MDs) simulation can provide more detailed information about both the microstructural properties and diffusion mechanisms at the atom level to develop functional materials (Figure 1). The results of MDS found that in sodium silicate, the extent of polymerization reduces from pure SiO_2_ to 2Na_2_O∙SiO_2_ [9,10,11,12]. Alongside the BO/NBO ratio and polymerization extent, other factors such as the bond length, bond angle, coordination number, distribution of ring sizes, and distribution of void sizes are also utilized to examine the structure at short- and mid-range scales. However, the spatial distribution of Na in Na_2_O∙3SiO_2_ remains not yet fully clarified. 

The very fast diffusivity of alkali atoms is one of the important dynamical properties of alkali silicates [13,14,15,16,17,18,19]. The addition of doped Na ions to pure SiO_2_ leads to a decoupling of alkali diffusion and diffusive transport in the Si-O network. Davidenko et al. suggested that a distribution where increasing alkali oxide content causes homogeneous, increasing disruption to the Si-O network of pure SiO_2_ is in conflict with the highly nonlinear dependence of viscosity on alkali concentration [20]. In accordance with other studies [21,22,23,24], the existence of the decoupling of alkali diffusion and diffusive transport in the relatively immobile Si-O network is proposed to interpret the fast diffusivity of alkali ions. The pre-peak at 0.9 Å^−1^ in the structure factor measured experimentally for alkali silicates is evidence for the diffusion pathway.

Various experimental results have shown that the structure of these silicates is found to comprise micro-regions with high Na concentrations. The two structural samples, the modified random network and the compensated continuous random network, predict some clustering of alkali atoms in the silicate’s structure [25,26,27]. This indicates that the clustering of alkali atoms marks out their diffusion pathway.

Several models for diffusion mechanisms in alkali silicate are suggested [28,29,30]. It is demonstrated that the diffusion pathway occupies a relatively small subspace of the system. According to Horbach et al. and Smith et al., the activated hopping of Na through the Si-O matrix is frozen with respect to the movement of Na atoms [9,31]. Based on the position of the first peak of the pair radial distribution function (RDF) for the Na-Na pair, they determine the mean hopping distance of Na atoms. The alkali ions hop into empty sites so that the mechanism owes more in character to their vacancy crystalline counterpart than to their interstitial cousins [32,33,34]. Habasaki et al. indicated that the cooperative motion mechanism may be suitable to clarify the increasing diffusivity with increasing alkali content [35]. Traps due to defects or impurities can lead to a decrease in charge carrier mobility and an increase in recombination rates in semiconductors [36]. However, many fundamental aspects of the fast diffusion of Na in Na_2_O∙3SiO_2_ remain up for debate.

Therefore, in this present study, we focus on Na_2_O∙3SiO_2_ at different temperatures. Based on the characteristics of the pair RDF, the Voronoi Si- and O-polyhedrons, the cluster function, *f_CL_(r)*, and the sets of fastest (SFA) and slowest atoms (SSA), we attempt to gain insight into the spatial distribution as well as the diffusion pathway of Na in the Na_2_O∙3SiO_2_.

## 2. Results and Discussion

### 2.1. Characteristics of Structural Na_2_O∙3SiO_2_

The total RDF for neutron diffraction, G(r), for Na_2_O∙3SiO_2_ is presented in Figure 2. This RDF exhibits the short-range order (SRO). As shown in Figure 2, the first peak of G(r) is located at 1.60 ± 0.02 Å, which is contributed to by *g_Si-O_(r)*, exhibiting the Si-O bond length. The second and third peaks of G(r) are located at 2.25 ± 0.02 Å and 2.60 ± 0.02 Å, which are contributed to by *g_Na-O_(r)* and *g_O-O_(r)*, exhibiting the Na-O and O-O bond lengths, respectively. It can be seen that there is good agreement between the experimental and simulated G(r) RDFs for an *r* of up to 6 Å [1]. Using the method in [37], the calculation result for the bond angle distribution indicated that the peaks of the O-Si-O and Si-O-Si bond angle distributions are located at 109.6° and 149.0°, respectively. For Si-O-Na, the bond angle distribution is centered around 90°–125°, very close to the experiments [1,7].

Figure 3 shows the RDF for the BO-Na, NBF-Na, O-Na, and Si-Na pairs at temperatures of 300, 973, and 1573 K. A pronounced peak is seen in *g_NBF-Na_(r),* which is not unclear for *g*_BO-Na_(*r*). Note that
(1)$$g_{O-Na}(r)=C_{BO}g_{Na-BO}(r)+C_{NBF}g_{Na-NBF}(r),$$
where $C_{BO}=m_{BO}/m_{O},C_{NBF}=m_{NBF}/m_{O},$ and *m_NBO_*, *m_BO_*, and *m_O_* are the total numbers of NBO, BO and O, respectively. This means that most of the Na is located around the NBF and is rarely present in the vicinity of BO. All the constructed samples consist of SiO_4_ along with BO and only fewer SiO_5_ particles in the high-temperature samples, indicating the tetrahedral network structure of Na_2_O∙3SiO_2_. The major O forms are NBO and BO, and less than 0.05% of the total O is FO. In addition, the relative fraction of O types was almost unchanged with temperature (Table 1). The spatial distribution of *T*O*y* (*T* is Si or Na, *y* = 4.5) in Na_2_O∙3SiO_2_ samples at different temperatures is shown. As seen, the structure of Na_2_O∙3SiO_2_ comprises mainly SiO_4_ units and a small count of NaO_4_ and NaO_5_ units, which are distributed over the whole space.

The fractions of different types of O atoms are listed in Table 1. It can be seen that O-polyhedrons either do not contain or do contain Na; meanwhile, Si-polyhedrons do not contain any Na residues. Our simulation shows that most O-polyhedrons comprise BO0, BO1, NBF0, NBF1, and NBF2. It shows that BO0 and NBF0 represent an empty O-polyhedron which does not contain Na. As temperature increases, the fractions *m*_BO1_/*m*_O_ and *m*_NBF0_/*m*_O_ increase. In contrast, *m*_BO0_/*m*_O_ and *m*_NBF2_/*m*_O_ slightly change in the opposite direction. From these data, we can suggest a simple diffusion sample in Na_2_O∙3SiO_2_. Namely, Na travels from its site to empty sites which are located in BO- and NBF-polyhedrons. The motion of Na between polyhedrons leads to the diffusivity of Na.

Next, more detail about the tetrahedral network structure of Na_2_O∙3SiO_2_ is derived from the characteristics of Voronoi polyhedrons, which are summarized in Table 2. As seen, the average volume per polyhedron increases in the order Si- → BO- → NBF-polyhedron. The total volume of BO- and NBF-polyhedrons, which contain Na, is about 87% of the simulation box. Although *V*_BO_ is equal to 1.6 times *V*_NBF_, about 78–90% of total Na is placed in NBF-polyhedrons. As temperature increases, the amount of Na residing in NBF-polyhedrons decreases from 91 to 78%, corresponding with the temperature increasing from 300 to 1573 K. This demonstrates that Na atoms are not uniformly distributed through O-polyhedrons but instead are gathered in NBF-polyhedrons, as can be seen from Figure 4. This supports the idea that the diffusion pathway of Na occurs in NBF-polyhedrons with a total volume of 33.0–33.6% of the simulation box.

Figure 5 plots the number of BO1-, NBF1-, and NBF2-polyhedrons versus time at different temperatures. As seen, the number of NBF2-polyhedrons is larger than that of BO1 polyhedrons at low temperatures, and it becomes smaller at high temperatures. In the temperature range of 773–973 K, the number of NBF2-polyhedrons is the same as that of BO1 polyhedrons. This is explained by the fact that more Na spreads on BO-polyhedrons at high temperatures. This is also observed from the spatial distribution of all types of O*T*_2_ linkages (*T* is Si or Na), as plotted in Figure 6. As seen, the Si-O-Na and Si-O-Si linkages are dominant. There is only a small count of Na-O-Na linkages. These linkages slightly change with increasing temperature. This is explained by the fact that pure SiO_2_ is composed of a continuous random network of SiO_4_ tetrahedra and that the doped Na ions break the Si-O linkages, which leads to the generation of non-bridging oxygen (NBO) in Na_2_O∙3SiO_2_. The generation of NBO lowers the glass melting point.

### 2.2. The Diffusion Pathway of Na Atoms in Na_2_O∙3SiO_2_

To demonstrate the clustering of Na atoms, we use the link cluster function, *f_Cl_*(*r*); the calculation algorithm employed here can be found elsewhere [38,39]. The sets of fastest, slowest, and random atoms (SFA, SSA, and SRA) comprise about 10% of all the atoms. The SFA has a mean square displacement (MSD) larger than that of the remaining atoms. The SSA has an MSD smaller than that of the remaining atoms. The SRA contains atoms that are randomly chosen from the sample. The atoms in the SFA, SSA, and SRA are determined from the atom position in the configuration at 150 ps. Figure 7 plots *f_Cl_*(*r*) at different temperatures. As seen, in the temperature range of 300–1573 K, the considered sets are mostly unchanged. However, the variations in the SSA, SFA, and SRA are quite different with *r*; namely, the SSA drops drastically from 1000 to 483 atoms as *r* varies from 1.3 to 1.75 ± 0.05 Å. The value of *f_Cl_*(*r*) at 1.75 Å is about 1000 and 895, corresponding to the SFA and SRA, respectively.

Clearly, these values are significantly larger than that of the SSA. With further increasing *r* to 3.00 ± 0.05 Å, the SFA and SRA appear as turning points, and then *f_Cl_*(*r*) gradually decreases. The result demonstrates the heterogeneous spatial distribution of the fastest and slowest atoms in the Na_2_O∙3SiO_2_ network. The problem here is that we do not know whether the Na atoms are mainly distributed in the SFA. To investigate this, we considered the distribution of sets of the fastest and slowest atoms using a visual technique. Here, we considered two consecutive configurations at moments *t* and *t* + 2 ps. Then, the MSD was identified, and the SFA and SSA were therefore found. Figure 8 displays the distribution of SFA and SSA at 773, 1173, and 1573 K. As can be seen, the distributions of both the fastest and slowest atoms are not uniform. The SFA mainly includes Na atoms, while the SSA mainly includes Si and O atoms. This demonstrates that the dynamics of the atoms are heterogeneous, and the hopping of Na atoms is mainly distributed for diffusion pathways in Na_2_O∙3SiO_2_.

To investigate the diffusion pathway, we specified the number of BO- and NBF-polyhedrons as a function of <*x*_A_>, where <*x*_A_> is the mean number of Na atoms in the A-polyhedron for the time *t*_sot_. The distributions of the fraction *m*_Ax_/*m*_A_ and the time dependence of the deviation of those distributions, *δ*_Ax_, for the case *t*_sot_ = 150 ps are plotted in Figure 9 and Figure 10. It can be seen that the curve for the NBF-polyhedrons possesses a pronounced peak at <*x*_A_> = 0.75 and spreads over a wide range, whereas the major BO-polyhedrons possess a small <*x*_A_>. As temperature increases, the curve for NBF spreads over a narrower range. This demonstrates that the diffusion pathway is composed of NBF-polyhedrons. With increasing *t*_sot_, the deviation, *δ*_Ax_, decreases quickly, and it approaches a smaller value at higher temperatures. This observation is understood as follows: under *t*_sot_, the average number of Na atoms in the *i*^th^ A-polyhedron approaches <*x*_Ai_>, which is proportional to
(2)$$<x_{Ai}>∼\exp(E_{Si}/k_{B}T),$$
where *E*_Si_, *k_B_*, and *T* are the site energy, Boltzmann constant, and temperature, respectively.

On the other hand, from Figure 9, the <*x*_A_> of BO is significantly smaller than that of NBF. This means that the site energy for a Na atom located in an NBF-polyhedron must be significantly smaller than that for a Na atom in a BO-polyhedron. In addition, <*x*_A_> varies over a wide range, indicating that Na has various energies, *E*_Si_. In fact, Na moves frequently between A-polyhedrons, so the average number of Na atoms in the *i*th polyhedron quickly approaches <*x*_Ai_>, and *δ*_Ax_ also decreases quickly with *t*_sot_. As temperature increases, the values of <*x*_Ai_> for different A-polyhedrons are close to each other. This leads to *δ*_Ax_ approaching a smaller value with increasing temperature (see Figure 10). Figure 11 displays the number of Na atoms staying in an A-polyhedron or moving from an A-polyhedron to another one within 2.0 ps. As expected, the number of Na atoms remaining monotonously decreases with increasing temperature.

Due to the movement of Na between O-polyhedrons, the system undergoes numerous Ax → Ax’ transformations. The average number of Ax → Ax’ transformations is plotted in Figure 10. As seen in Figure 10, there are a small number of BOx atoms undergoing the transformation BO0 ↔ BO1, which increases with temperature. Therefore, more Na atoms reside in BO-polyhedrons at higher temperatures, and Na atoms perform independent jumping in them. In the case of NBF-polyhedrons, the transformations NBF0 ↔ NBF1 and NBF1 ↔ NBF2 occur in the majority of polyhedrons. However, the system comprises a number of NBF-polyhedrons undergoing the transformation NBFx → NBFx’, where |*x* − *x’*| > 1. The number of such NBF-polyhedrons also increases with increasing temperature. We conclude that, unlike BO-polyhedrons, Na performs independent jumping and cooperative motion. Cooperative motion is realized in more NBF-polyhedrons at higher temperatures.

From the above explanation, we suggest that the diffusion constant for Na in Na_2_O∙3SiO_2_ may be written as follows:(3)$$D_{Na}=\gamma f<d_{SS}^{2}>v_{p},$$
where $<d_{SS}^{2}>$ is the mean square distance between a site and its nearest neighbor, $v_{p}$ is the rate of Na atoms moving between O-polyhedrons, γ is the geometrical factor, and *f* is the correlation coefficient describing the forward–backward jumps of Na atoms [32,33].

## 3. Materials and Methods 

The simulation was carried out for Na_2_O∙3SiO_2_ at ambient pressure over a temperature range of 300–1573 K. The sample was made of 9996 atoms, including 5831 O atoms, 2499 Si atoms, and 1666 Na atoms. All simulation runs were performed using MXDORTO code [40]. We used the interaction potentials, consisting of two-body and three-body terms, which can quite reliably reproduce the structure and dynamics of sodium silicate. The density was adopted from that of a real sodium trisilicate glass of 2.4323 g/cm^3^ [41]. A complete description of these potentials can be found elsewhere [29,42]. The pair potential has the following form:(4)$$U_{ij}(r_{ij})=\frac{Z_{i}Z_{j}e^{2}}{4\pi \varepsilon_{0}r_{ij}}+f_{0}(b_{i}+b_{j})\exp\left(\frac{a_{i}+a_{j}-r_{ij}}{b_{i}+b_{j}}\right)+\frac{c_{i}c_{j}}{r_{ij}^{6}}+D_{1ij}\exp(-\beta_{1ij}r_{ij})+D_{2ij}\exp(-\beta_{2ij}r_{ij})$$

The potential parameters are listed in Table 3. Note that the three-body term relating to the Si-O-Si angle has the following form:(5)$$U_{ijk}=-f\left[\cos\{2(\theta_{kij}-\theta_{0})\}-1\right]\sqrt{k_{ij}k_{jk}}$$
(6)$$k_{ij}=\frac{1}{\exp[g_{r}(r_{ij}-r_{m})-1}$$
where *f* is the force constant; *θ_kij_* is the angle among atoms *k–i–j*; and *θ*_0_, *g_r_*, and *r_m_* are the parameters for adjusting the angular part of covalent bonds. The partial charges for the Si and O atoms are calculated as follows: Z_Si_.Z_O_ = −2.5 and 2Z_Na_ + 3 Z_Si_ + 7 Z_O_ = 0, where Z_Na_ is fixed at 1.0.

The pair radial distribution function (PRDF) for BO and NBF is characteristic of their local microstructure. Here, BO, NBO, and FO are the types of oxygen which are bounded, respectively, with two, one, and no Si atoms; NBF is denoted either as NBO or FO. Overall, the status of O (BO, NBO, and FO) was mostly unchanged during the simulation. Only a few BO ↔ NBF transformations were detected at temperatures of 300, 973, and 1373 K.

In this work, the simple nanoarchitectonics of atoms include O-polyhedrons and Si-polyhedrons; bridging oxygen (BO)-, non-bridging oxygen (NBO)-, and free oxygen (FO)-polyhedrons; the group of Na in NBO- and FO-polyhedrons; and the fastest, slowest, and random atoms. The visualization of MD data was carried out to study the structural properties and diffusion pathway of Na atoms in sodium trisilicate [43]. In this context, we suppose that the simulation box is fully filled by O- and Si-polyhedrons, and Na is placed inside these polyhedrons. In this work, the A-polyhedron is denoted as the A-centered polyhedron, where A is the Si, BO, or NBF; we also use the Ax-polyhedron, where *x* is the number of Na atoms in the A-polyhedron. For instance, NBF1 represents the NBF-polyhedron containing one Na atom, respectively. During the simulation process, we found that Na atoms were not placed in fixed polyhedrons, but they frequently moved from one to other polyhedrons. To clarify this effect, we observed A-polyhedrons in configurations separated by 2 ps. The local Na density in the vicinity of an O or Si atom was quantified by the mean number of Na atoms in the A-polyhedron, which is called <*x*_A_>. We determined Ax for configurations within a span of time *t*_sot_, and then <*x*_A_> was obtained by averaging *x* over those polyhedrons. The value of <*x*_A_> depends on *t*_sot_ and approaches a finite value as *t*_sot_ → ∞. Consider two consecutive configurations at moments *t* and *t* + 2 ps. We specified the number of Na atoms staying in the A-polyhedron and also the number of Na atoms moving to other polyhedrons. In this way, we detected Ax → Ax’ transformations. For instance, BO1 at moment *t* transforms to BO0 at moment *t* + 2 ps if one Na atom leaves the given BO-polyhedron to enter another O-polyhedron. The number of Ax → Ax’ transformations characterizes sodium’s migration in sodium silicate. To investigate the microstructure at an atomic level, MATLAB2018 software, code by N.V. Hong (Hanoi University of Science and Technology, VIET NAM) was used to visualize the MD data in 3D space.

## 4. Conclusions

In this study, using nanoarchitectonics and an MD simulation, the structural properties and diffusion pathway of alkali in sodium silicate (Na_2_O∙3SiO_2_) were considered. The obtained results suggest that most Na atoms are located around NBF. Si-polyhedrons do not contain Na, and O-polyhedrons either are empty or contain Na atoms. This demonstrates that Na atoms are concentrated in a relatively small subspace. The motion of Na atoms includes the displacement between two neighboring NBF-polyhedrons, which describes the channel for Na diffusion. In addition, the doped Na ions break the Si-O linkages, which leads to the generation of non-bridging oxygen (NBO) in Na_2_O∙3SiO_2_.

Na_2_O∙3SiO_2_ undergoes numerous transformations, such as NBF0 ↔ NBF1, NBF1 ↔ NBF2, and BO0 ↔ BO1. The dynamics in Na_2_O∙3SiO_2_ are mainly distributed through the hopping of Na atoms; namely, Na atoms carry out independent hopping when transformations occur. We propose that the diffusion of Na atoms in Na_2_O∙3SiO_2_ is realized through independent hopping in BO- and NBF-polyhedrons and through cooperative motion in NBF-polyhedrons.

This new diffusion model is proposed to give insight into the diffusion pathway of alkali in sodium silicate systems. These results provide a foundation for future experimental research aimed at fabricating materials for industrial applications, ceramics, metallurgy, and glass technologies, as well as understanding the fundamentals of minerals. The findings revealed in this study are expected to contribute to future studies on new materials with varying temperature and pressure conditions.

## Figures and Tables

**Figure 1 ijms-25-05628-f001:**
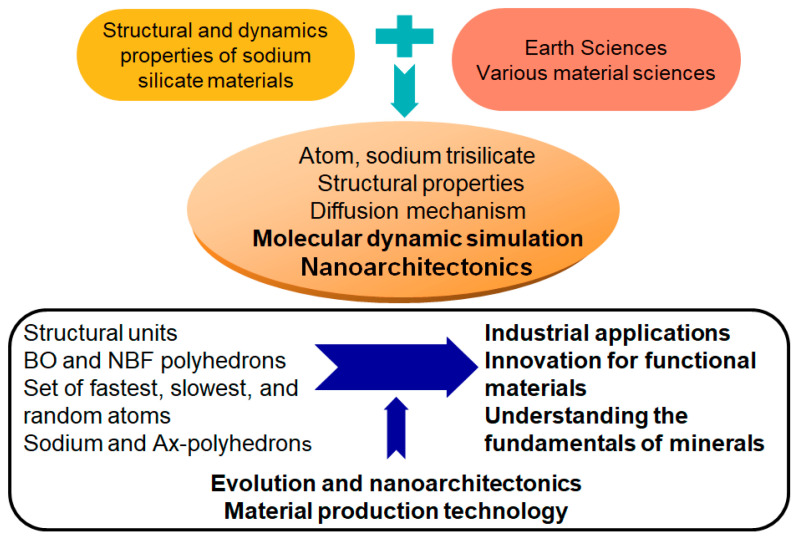
Outline of nanoarchitectonics and molecular dynamics simulation: meaning and procedure.

**Figure 2 ijms-25-05628-f002:**
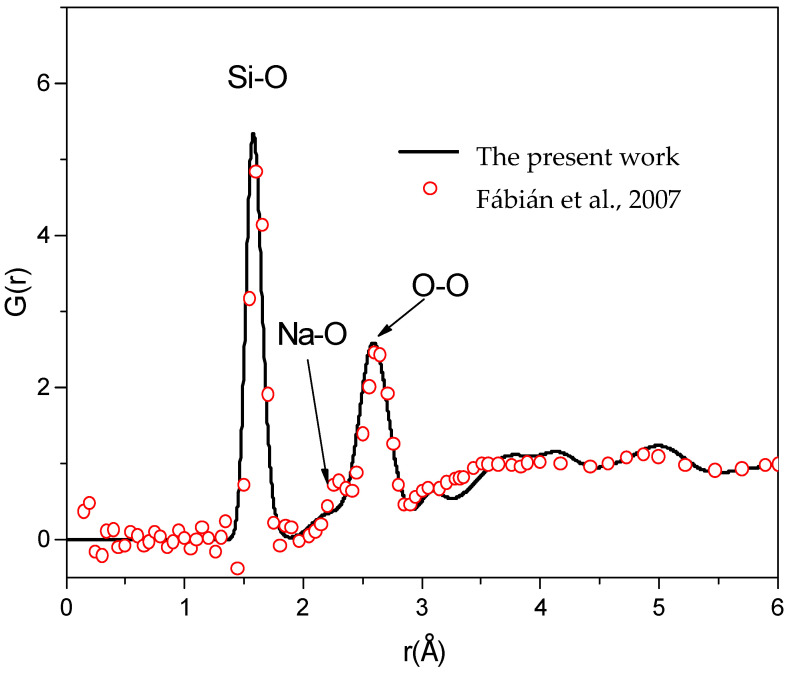
The total RDF of sodium trisilicate at a temperature of 973 K and a comparison with experimental data [1], which was obtained from this work and reported in an experiment in [1].

**Figure 3 ijms-25-05628-f003:**
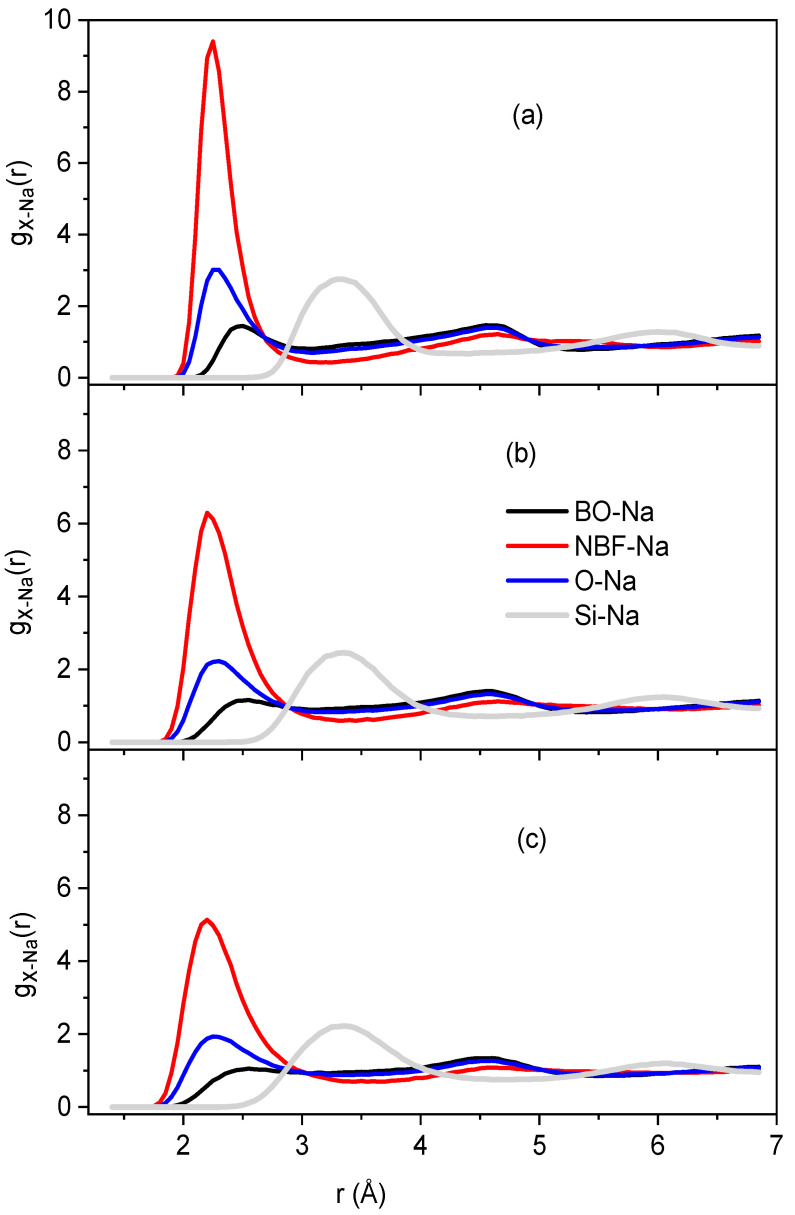
The pair RDF for BO-Na, NBF-Na, O-Na, and Si-Na pairs at different temperatures: (**a**) 300 K; (**b**) 973 K, and (**c**) 1573 K.

**Figure 4 ijms-25-05628-f004:**
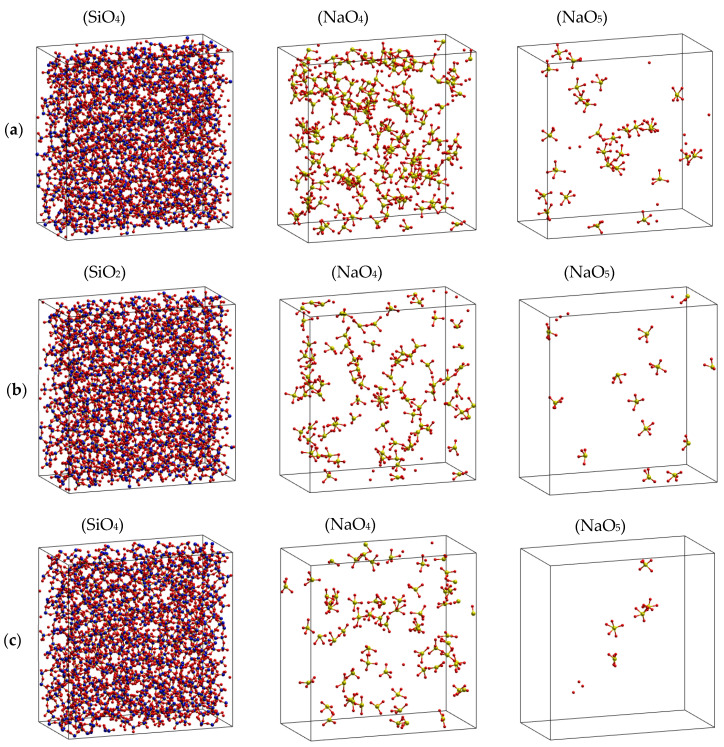
Spatial distribution of *T*O*y* (*T* is Si or Na, *y* = 4.5) in Na_2_O-3SiO_2_ models at different temperatures. Here, (**a**) 300 K, (**b**) 973 K, and (**c**) 1573 K, with O (red color), Si (blue color), and Na (yellow color).

**Figure 5 ijms-25-05628-f005:**
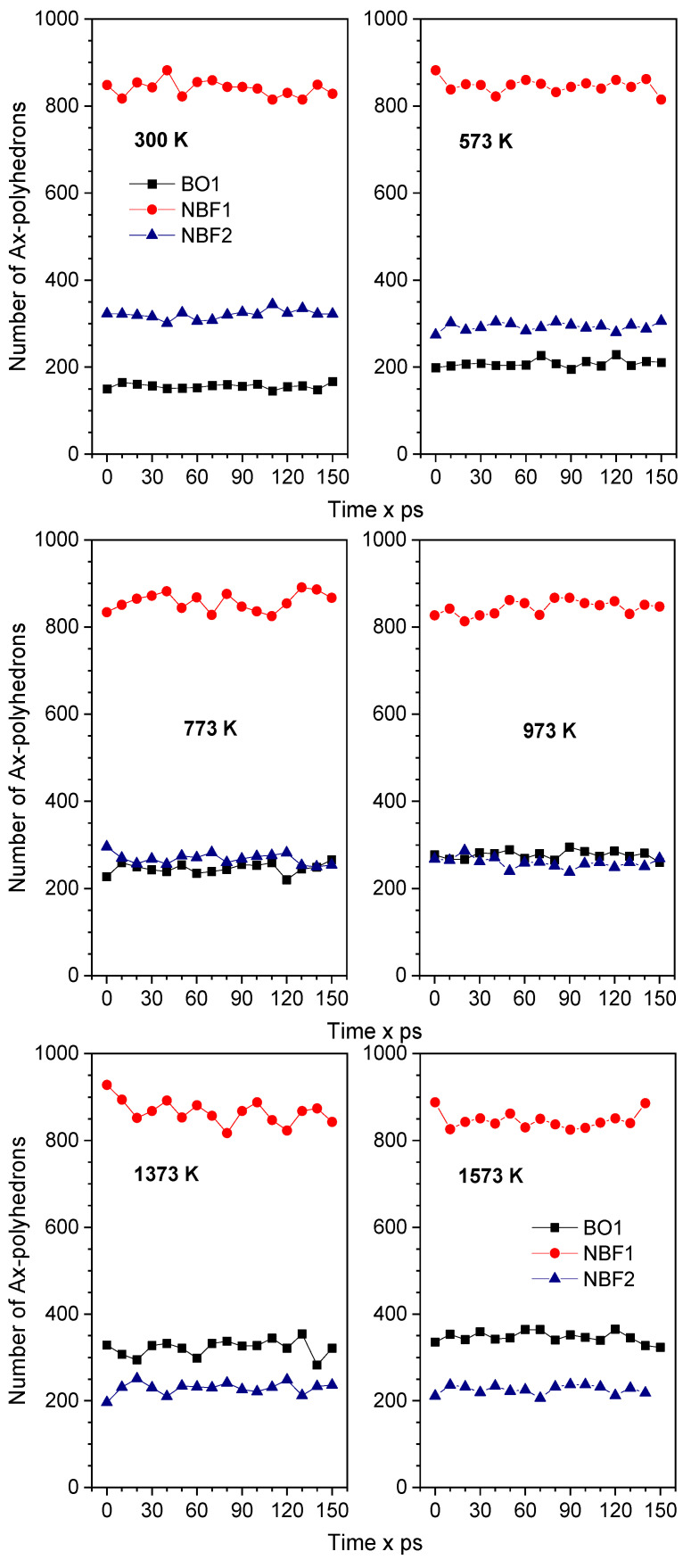
Time dependence of the number of Ax-polyhedrons at different temperatures. Here, A denotes an NBF or BO atom; *x* denotes the number of Na atoms in the Ax-polyhedron.

**Figure 6 ijms-25-05628-f006:**
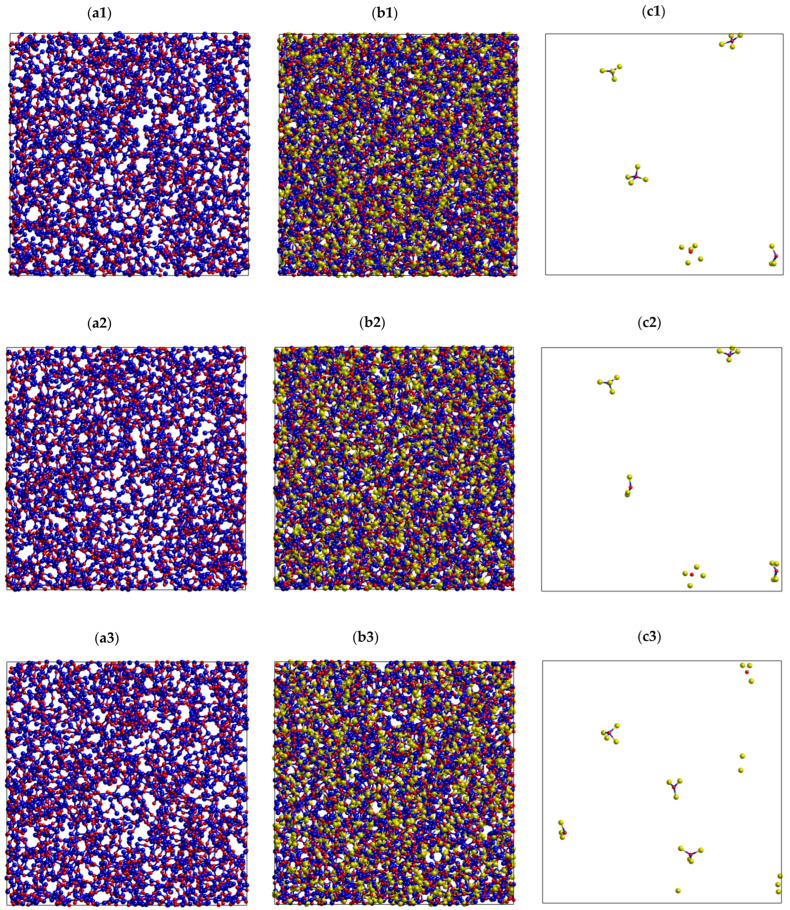
Spatial distribution of all types of O*T*_2_ linkages (*T* is Si or Na) in Na_2_O-3SiO_2_ at different temperatures, (**a1**–**c1**) 300 K; (**a2**–**c2**) 973 K; and (**a3**–**c3**) 1573 K, with O (red color), Si (blue color), and Na (yellow color). Here, (**a1**–**a3**) is Si-O-Si; (**b1**–**b3**) is Si-O-Na; and (**c1**–**c3**) is Na-O-Na linkage, respectively.

**Figure 7 ijms-25-05628-f007:**
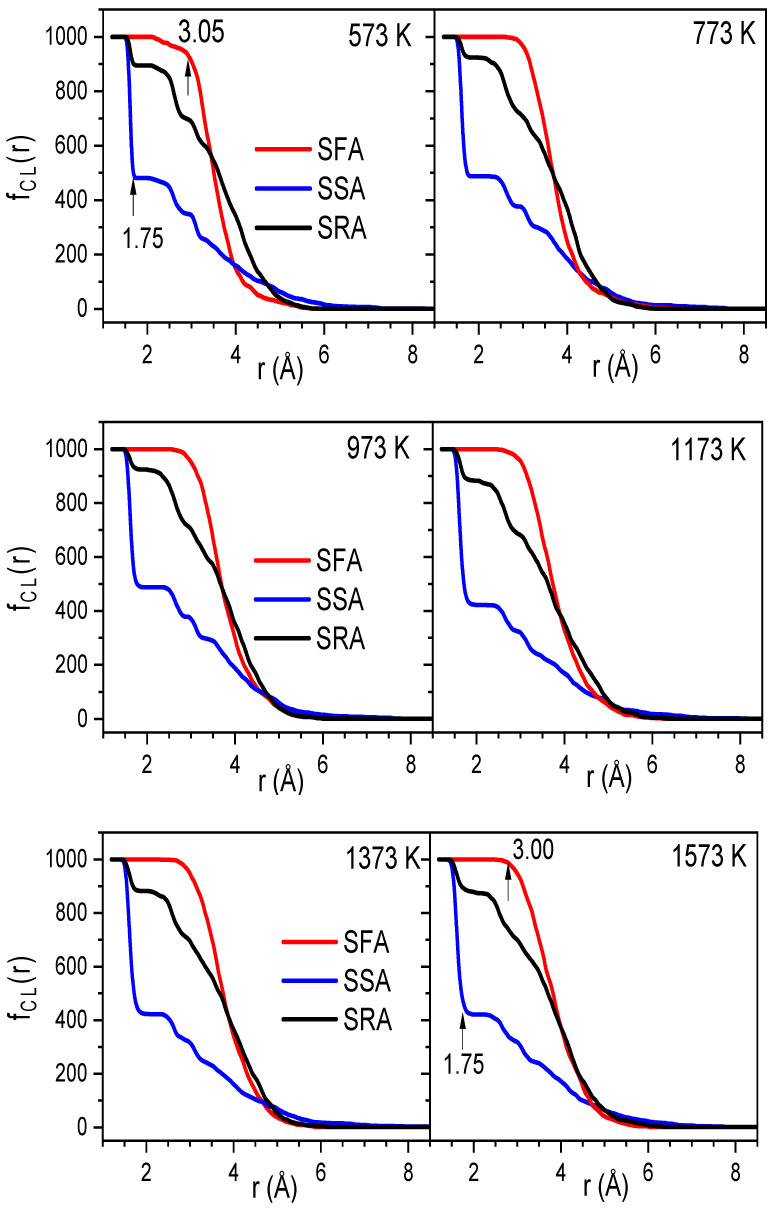
The CL function for atoms belonging the SFA, SSA, and SRA at different temperatures.

**Figure 8 ijms-25-05628-f008:**
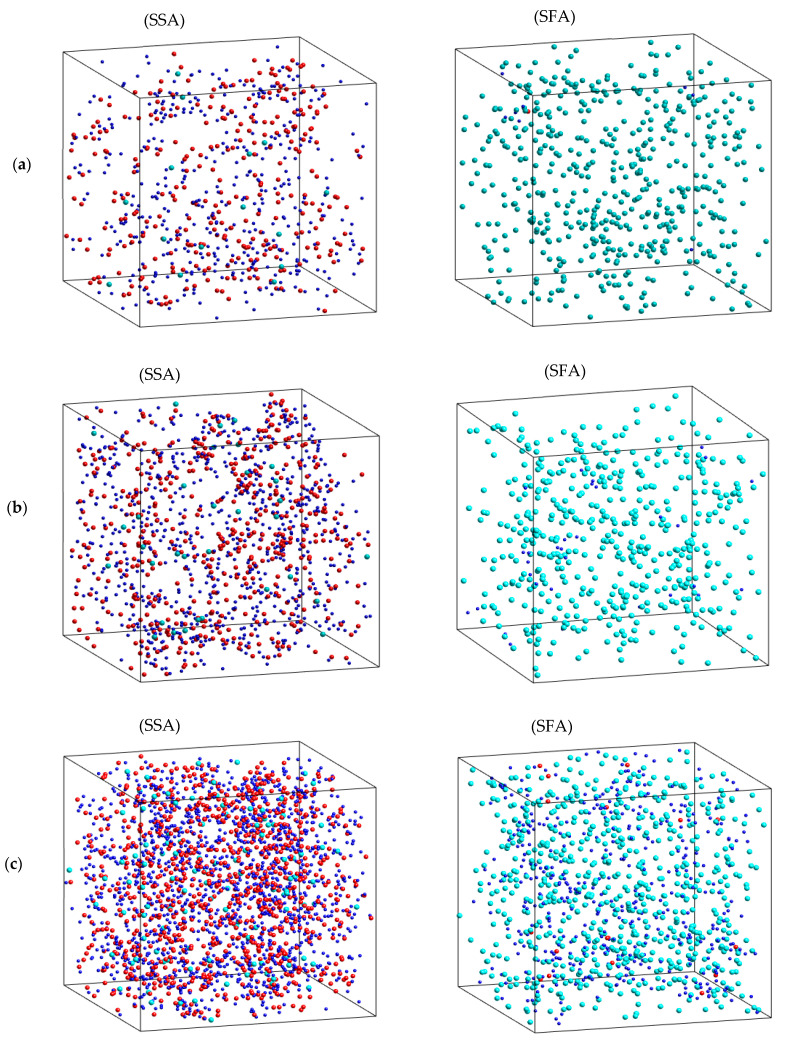
The 5% distribution of the sets of slowest atoms (SSA) and fastest atoms (SFA): (**a**) 1573 K, (**b**) 1173 K, and (**c**) 773 K; the red ball is Si, the blue ball is O, and the green ball is Na.

**Figure 9 ijms-25-05628-f009:**
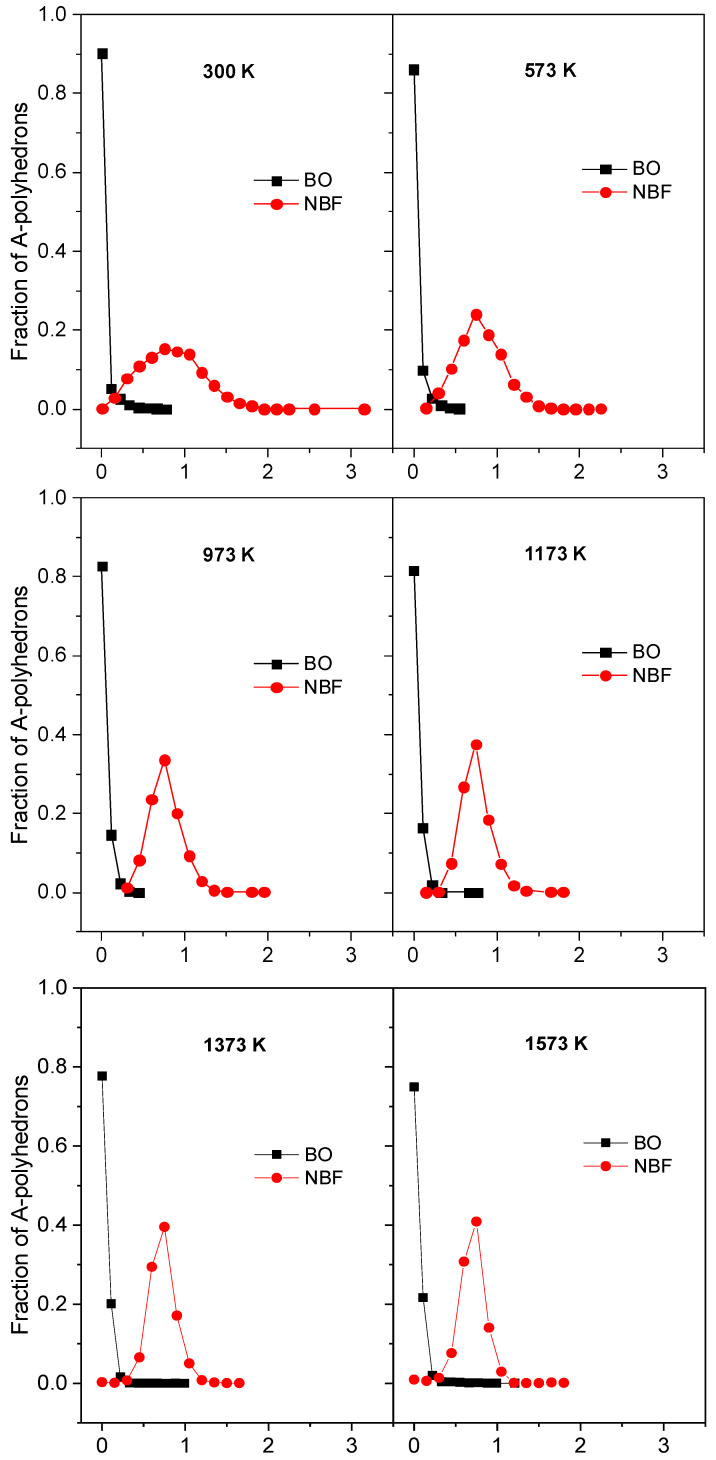
The fraction *m*_Ax_/*m*_A_ as a function of the average number of Na atoms in the A-polyhedron <*x*_A_> for 150 ps at different temperatures. Here, A denotes BO or NBF; *m*_Ax_ and *m*_A_ denote the number of A-polyhedrons with <*x*_A_> and the total number of A-polyhedrons, respectively.

**Figure 10 ijms-25-05628-f010:**
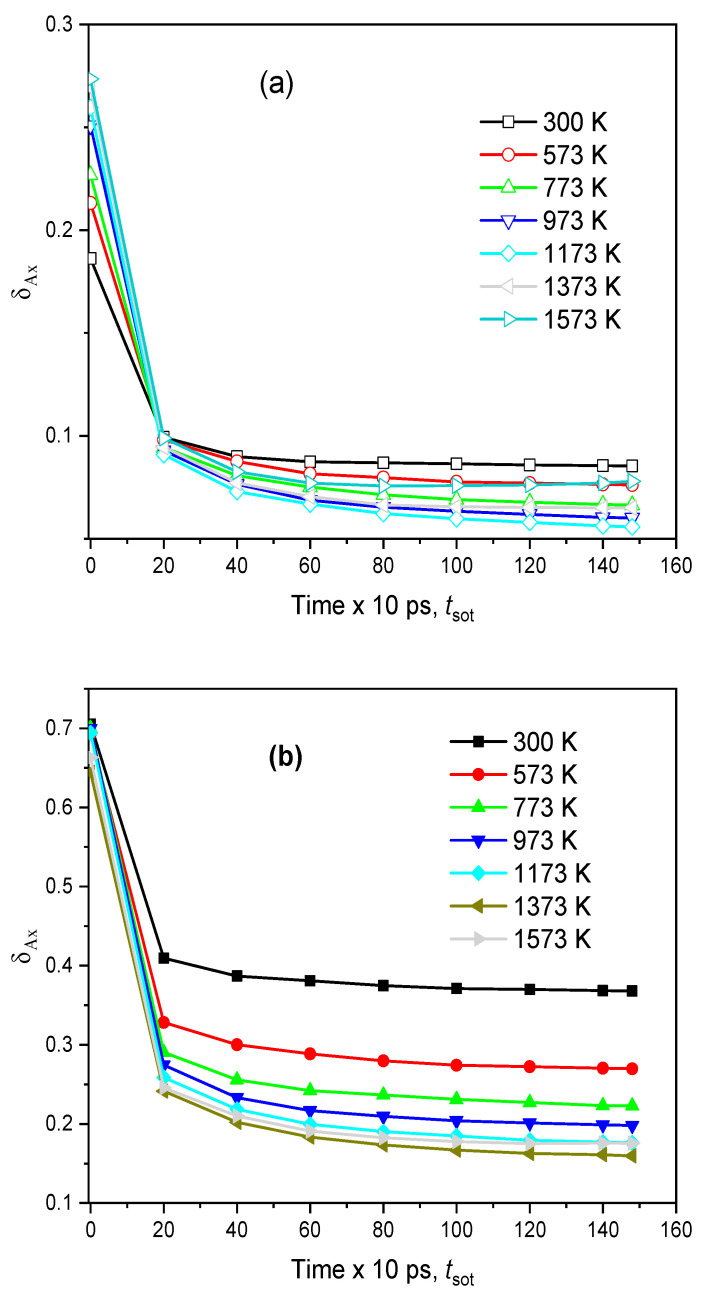
The time dependence of the deviation, *δ*_Ax_, at different temperatures: (**a**)—BO and (**b**)—NBF.

**Figure 11 ijms-25-05628-f011:**
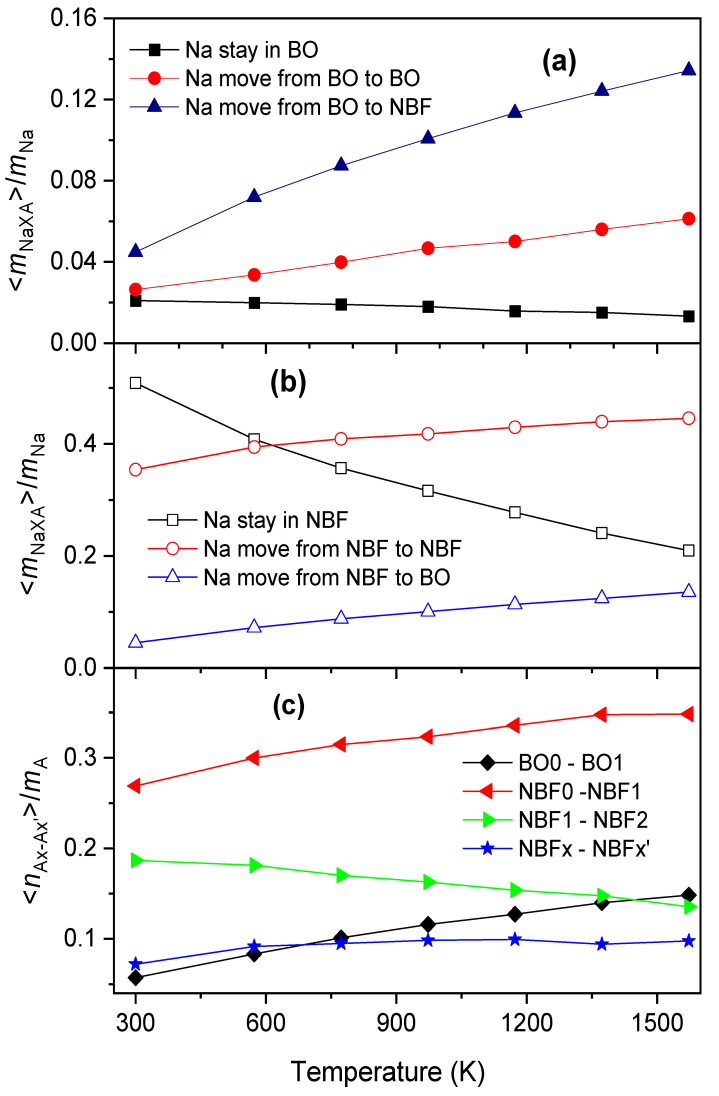
The temperature dependence of <*m*_NaA_>/*m*_Na_ (**a**,**b**) and <*m*_Ax-Ax’_>/*m*_A_ (**c**). Here, <*m*_NaA_> is the average number of Na atoms staying in A-polyhedrons or moving from an A-polyhedron to another one within 2 ps; A is BO or NBF; *m*_Na_ is the total number of Na atoms; <*m*_Ax-Ax’_> is the average number of Ax → Ax’ transformations within 2 ps; and *m*_A_ is the total number of A-polyhedrons.

**Table 1 ijms-25-05628-t001:** Distribution of Na in O-polyhedrons and fraction of different types of O. Here, *m_Ax_* is the number of Ax-polyhedrons; A is BO or NBF; x is the number of Na atoms in the Ax-polyhedron; *m_FO_*, *m_NBO_*, *m_BO_*, and *m_O_* are the total numbers of FO, NBO, BO, and O atoms, respectively.

T (K)	*m_BO_/m_O_*	*m_BO_* _0_ */m_O_*	*m_BO_* _1_ */m_O_*	*m_NBF_/m_O_*	*m_NBF_* _0_ */m_O_*	*m_NBF_* _1_ */m_O_*	*m_NBF_* _2_ */m_O_*
300	0.7151	0.6884	0.0268	0.2849	0.0842	0.1441	0.0550
573	0.7151	0.6794	0.0357	0.2849	0.0880	0.1452	0.0502
773	0.7151	0.6729	0.0422	0.2849	0.0903	0.1471	0.0460
973	0.7151	0.6676	0.0475	0.2849	0.0941	0.1448	0.0445
1173	0.7151	0.6640	0.0511	0.2849	0.0955	0.1454	0.0429
1373	0.7145	0.6591	0.0552	0.2855	0.0967	0.1485	0.0393
1573	0.7145	0.6591	0.0552	0.2855	0.0967	0.1485	0.0393

**Table 2 ijms-25-05628-t002:** Characteristics of Voronoi polyhedrons. <*v*_Si_>, <*v*_BO_>, and <*v*_NBF_> are the average volumes per Si-, BO-, and NBF-polyhedron, respectively, measured in Å^3^; *V*_Si_, *V*_BO_, *V*_NBF_, and *V*_SB_ are the volumes occupied by Si-, BO-, and NBF-polyhedrons and the volume of the simulation box, respectively; *m*_NaBO_ and *m*_NaNBF_ are the numbers of Na atoms residing in BO- and NBF-polyhedrons, respectively; and *m*_Na_ is the total number of Na atoms.

T (K)	<*v*_Si_>	<*v*_BO_>	<*v*_NBF_>	*V*_Si_/*V*_SB_	*V*_BO_/*V*_SB_	*V*_NBF_/*V*_SB_	*m*_NaBO_/*m*_Na_	*m*_NaNBF_/*m*_Na_
300	7.33	19.37	29.40	0.1237	0.5461	0.3302	0.0990	0.9010
573	7.43	19.51	29.72	0.1244	0.5446	0.3310	0.1218	0.8782
773	7.52	19.64	30.10	0.1246	0.5428	0.3326	0.1555	0.8445
973	7.61	19.75	30.47	0.1250	0.5420	0.3330	0.1603	0.8397
1173	7.70	19.93	30.88	0.1254	0.5417	0.3329	0.1771	0.8229
1373	7.82	20.11	31.41	0.1255	0.5380	0.3365	0.1843	0.8157
1573	7.94	20.37	31.96	0.1260	0.5374	0.3366	0.2119	0.7881

**Table 3 ijms-25-05628-t003:** The interatomic potential parameters.

Atomic Parameters	*Z*	*a* (nm)	*b* (nm)	*c* (kJ/mol∙nm^3^)
Si	-	0.099759	0.00830	0.000
O	-	0.181819	0.01539	27,400
Na	1.00000	0.139500	0.01150	10,000
Pair parameters	*D*_1_ (kJ/mol)	*β*_1_ (1/nm)	*D*_2_ (kJ/mol)	*β*_2_ (1/nm)
SiO	668,428	59.636	−105,335	45.514
3-body parameters	*f* (kJ/mol)	*θ*_0_ (degree)	*r_m_* (nm)	*g_r_* (1/nm)
Si-O-Si	0.0006	147.0	0.170	168.0

## Data Availability

The original contributions presented in the study are included in the articl, further inquiries can be directed to the corresponding author.

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
