# Peer review of "The Characteristics of Structural Properties and Diffusion Pathway of Alkali in Sodium Trisilicate: Nanoarchitectonics and Molecular Dynamic Simulation"

_ijms, 2024, doi:10.3390/ijms25115628_

Round 1

Reviewer 1 Report

Comments and Suggestions for Authors

This work researched investigate the structural properties and diffusion pathway of Na atoms in Na2O-doped SiO2 in a wide temperature range on the basis of molecular dynamics simulations. This manuscript actually provides interesting data. These data are valuable of being published in scientific journals such as Int. J. Mol. Sci. However, this manuscript was not prepared understandably for non-specialist readers where tastes of innovations and novelty are less. The latter features have to be fixed through revisions. Please see below.

1) This manuscript only shows research results as figures. Therefore, the outline cannot be instantly understandable for readers. It is better to add one initial figure to explain target systems and objectives of this research.

2) Some important keywords have to be well included in the title. In this case, words related with simulation, theory, or calculation have to be included. In addition, new conceptual term related with nanostructures had better be included to increase innovative features. I may suggest use of an emerging conceptual term, nanoarchitectonics, in the title (for the concept, see https://academic.oup.com/bcsj/article/97/1/uoad001/7457599). For example, the title like ... The characteristics of structural properties and diffusion pathway of alkali in Na2O-doped SiO2: nanoarchitectonics and simulation ... may sound more appropriate.

3) Descriptions on conclusion are not so impactful. Not limited to descriptions of the results summary, more descriptions on impacts of this research to more practical applications and related sciences had better be added. Future perspective descriptions are necessary for conclusion part.

Author Response

Dear Reviewer 1,

After carefully studying the reviewer's comments and suggestions for the manuscript, we revised the major part of the manuscript as follows

Best regards,

Authors

Reviewer 2 Report

Comments and Suggestions for Authors

The submitted manuscript entitled “The characteristics of structural properties and diffusion pathway of alkali in Na2O-doped SiO2” by Trang etal. The reported sodium-doped silicate glass is a model for multicomponent silicate glass that has many diverse applications in the glass and electronics industries. In the submitted work, the authors were trying to reveal the role played by the alkali ions through molecular dynamics simulation studies. The studies concluded that the doped alkali ions form weak bonds, which changes the local environment of the Si atoms in the network.  This contribution may bring new insights into the fundamental understanding of the electronic properties of alkali silicate glass and their movements/diffusion. The work is well justified, however, the work in its present form is not suitable and needs some revisions before rendering a final decision.

My specific points are below

·         Please provide the structural parameters for simulated glass structures.

·         Have you calculated the distribution of bond angle (like Si-O-Na) ?

·         What is the band gap for the chosen material for different dopant levels?

·         Figure 1 caption is unclear.

·         Better not to use the word “references” often in the manuscript text. Just quoting the numbers will do.

·         First like in the introduction, what is the second word “It” The It is well-known?

·         All the chemical formulas must be checked and formatted appropriately with the number in the subscript.

·         The electronic properties and diffusion of ions can be referred to one of the recent similar work reported in the literature doi.org/10.1021/acsami.1c16287.

·         Make sure that the stated references 39 references are embedded in the text.

Author Response

Dear Reviewer 2,

After carefully studying the reviewer's comments and suggestions for the manuscript, we revised the major part of the manuscript as follows.

Best regards,

Authors

Reviewer 3 Report

Comments and Suggestions for Authors

The research motivation is good, but the quality of writing and presentation is poor. One can readily understand the latter upon reading the abstract of the paper. Moreover, since the short-hand notation such as MDS was not appeared more than once, it is not necessary to use such notations in abstract. I have significant difficulty to understand the language, and thereby, I urge the authors to rewrite the paper with the help of a native writer. My other comments are as follows:

1.      The first line of the Introduction reads “The It is well known that the structure of silica (SiO2) is an archetypal ..” I am not sure why it is “The  It is …”

2.      The last paragraph of the study does not clarify the objective of the study. What is the key interest? What these authors want to study and why?

3.      I cannot see names of software used in this study.

4.      The caption of Fig. 1 reads “Fig. 1 The neutron total RDF for Na silicate obtained from this work and reported in experiment [38].” However, the text on the Figure refers to reference [1]. I am not sure whether it is reference [38] or [1]?

5.      Fig. 7 does not clarify whether atoms are dispersed, or bonded? I see it is the former.

There are many other similar issues, which I do not write as of now. The paper needs rewriting, resubmission and further review.  

Comments on the Quality of English Language

-

Author Response

Dear Reviewer 3,

After carefully studying the reviewer's comments and suggestions for the manuscript, we revised the major part of the manuscript as follows.

Best regards,

Authors

Round 2

Reviewer 2 Report

Comments and Suggestions for Authors

In this reviewer's opinion, the revised version is suitable to render a final decision.

Reviewer 3 Report

Comments and Suggestions for Authors

Authors have revised their paper based on my suggestion. It may now be considered for possible publication. However, authors should carefully read their paper once again to remove typo related errors

Comments on the Quality of English Language

--